# Chernobyl's Lesser Known Design Flaw: The Chernobyl Liquidator Medal—An Educational Essay

**Michael McIntire \* and John Luczaj** 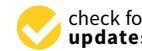

Department of Natural & Applied Sciences, University of Wisconsin–Green Bay, Green Bay, WI 54311, USA
\* Correspondence: mcintirm@uwgb.edu; Tel.: +1-920-465-5131

**Abstract:** The honorary Chernobyl Liquidator Medal depicts pathways of alpha, gamma, and beta rays over a drop of blood, signifying the human health impacts of the Chernobyl accident. A relativistic analysis of the trajectories depicted on the Chernobyl Liquidator Medal is conducted assuming static uniform magnetic and electric fields. The parametric trajectories are determined using the energies of alpha ($\alpha$) and beta ($\beta$) particles relevant to the Chernobyl nuclear power plant accident and compared with the trajectories depicted on the liquidator medal. For minimum alpha particle velocity of $0.0512c$, the beta particle trajectory depicted on the medal is highly unlikely to have come from a naturally occurring nuclear decay process. The parametric equations are used to determine the necessary beta energies to reproduce the depicted trajectories. This article documents the unfortunate misrepresentation of a famous scientific experiment on an honorary medal and illustrates the importance of better communication between artists and scientists.

**Keywords:** Chernobyl; liquidator; medal; radiation; trajectories; physics; design

## 1. Introduction

### 1.1. The Chernobyl Power Station Accident and the Liquidators

With near universal acceptance of global climate change by today's scientific community, coupled with a looming energy shortage as carbon-based fuels become increasingly limited, there has been a revitalization of nuclear energy throughout much of the world. This renewed focus on nuclear power has led many to reweigh the pros and cons of nuclear power in increasingly public debates. One of the perceived roadblocks is the public perception of nuclear safety, which invariably brings to mind images of the Chernobyl disaster that occurred on 26 April 1986. The accident at the Chernobyl Nuclear Power Station in Ukraine was one of the world's worst nuclear accidents and is considered a defining moment in the history of nuclear energy. It resulted from a poorly executed electrical engineering experiment performed on a RBMK-style reactor, which resulted in the release of approximately $14 \times 10^{18}$ Becquerels of radiation (as of 26 April 1986 and including noble gases) [1]. The 20th anniversary of the accident brought worldwide attention to the human, ecological, and economic impacts resulting from the disaster that were evaluated as part of the Chernobyl Forum [2]. This year marks the 33th anniversary of the Chernobyl accident.

In addition to local inhabitants in the region surrounding the Chernobyl power station, cleanup workers, known as "liquidators", were one of the main populations exposed to significantly elevated levels of radiation from the Chernobyl accident. In all, about 600,000 people participated in cleanup, decontamination, and support activities. The liquidators received various types of recognition for their efforts, including the medal pictured in Figure 1. While developing a college course on radioactivity and the environment at our university, we have discovered a surprising discrepancy in the design of this medal that we have not found published elsewhere.

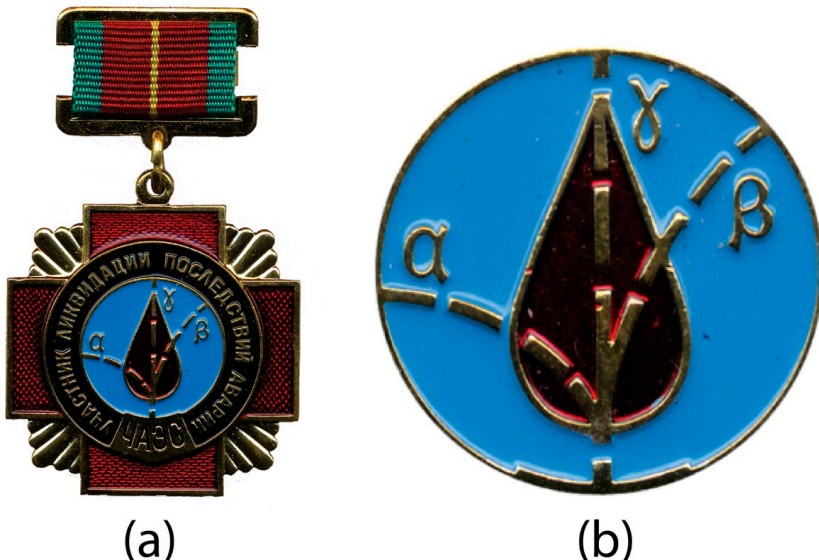

**Figure 1.** (**a**) Chernobyl Liquidator Medal (CLM) awarded to members of the Soviet Union for participation in cleanup activities after the reactor accident on 26 April, 1986. (**b**) A close-up of the central figure of the CLM showing the three radiation trajectories superimposed on a drop of blood. The diameter of the blue circular central figure is 1.8 cm.

More than a century ago, Ernest Rutherford demonstrated the effect that a magnetic field has on α, β, and γ radiation emitted from a sample of radium in a vacuum (Figure 2). His classic figure shows that for a strong uniform magnetic field perpendicular to and into the plane of the paper, α rays are slightly deflected to the left, β rays are deflected to a greater degree toward the right, and γ rays are not deflected at all [3]. A similar result has been demonstrated for an electric field between two electrodes, in which the α particles are deflected somewhat toward the negatively charged electrode, the β particles are deflected to a greater degree toward the positively charged electrode, and the γ rays are not affected. For α and β rays, the direction and magnitude of the deflection in both fields are dependent on the mass, charge, and velocity of the particles passing through the field. As illustrated in Rutherford's figure, the β particles vary significantly in their amount of deflection due to a greater variability in velocity compared to the α particles. Figures showing the nuclear decay trajectories have become classic illustrations and are reproduced in numerous physics and chemistry textbooks and on the World Wide Web [4–8].

The Chernobyl liquidator medal has a depiction similar to this classic diagram showing the deflection of nuclear decay products in a magnetic or electric field. Three types of radiation are superimposed on a drop of blood set over a blue background (Figure 1). A translation of the Russian words around the perimeter of the medal state: "Participant of the liquidation of the consequences of accidents." The three paths are shown emanating from the base of the medal near the four-letter abbreviation for the Chernobyl Atomic Electric Station.

To our surprise, it appears that there is something wrong with the representation included on the Chernobyl Liquidator Medal (CLM). While the direction of deflection does not particularly matter, since the electric charge or direction of a magnetic field is not indicated, a closer inspection reveals a larger problem. The deflection of the α particles on the CLM is significantly greater than for the β particles. This observation made us consider whether the CLM's depiction was actually possible, especially given that photographs of this medal are quite commonly shown to students, including in our university's course on radioactivity.

This article presents a detailed examination of the α, β, and γ radiation trajectories depicted on the famous Chernobyl Liquidator Medal that confirms the erroneous artistic rendering of Rutherford's

famous diagram. We would like to emphasize that our article intends no disrespect toward the residents or cleanup workers of the Chernobyl region or to the CLM artist.

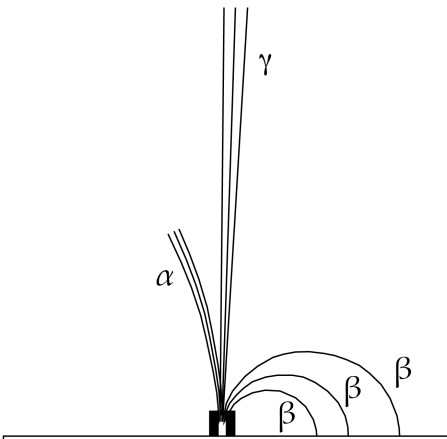

**Figure 2.** Lord Ernest Rutherford's classic diagram showing the deflection of α and β particles that emanate from a radioactive source into a magnetic field. Modified from [3].

### 1.2. Nuclear Decay Background

Radioactive nuclei are unstable structures that spontaneously decay into more stable nuclei by ejecting particles and electromagnetic radiation. The speed of the ejected particles depends on the mass and kinetic energy of the particle, which depends on the specific nucleus that is decaying. The kinetic energies of particles ejected during a nuclear decay can be converted into velocities using the relativistic equations [9],

$$E = mc^2 = \gamma m_0 c^2 \text{ and } \gamma = \frac{1}{\sqrt{1-\beta^2}} \tag{1}$$

where $\gamma$ is the relativistic gamma factor with $\beta = \frac{v}{c}$, $E$ represents the total energy of the ejected particle, $m_0$ is the rest mass, $v$ is the velocity, and $c$ is the speed of light. This yields

$$\beta = \sqrt{1 - \left(\frac{m_0 c^2}{m_0 c^2 + T}\right)} \tag{2}$$

where $T$ is the kinetic energy of the particle. While there are several decay mechanisms by which unstable nuclei relax, we will concentrate on gamma decay, beta decay, and alpha decay.

In gamma decay, the atomic nucleus undergoes a transition from a higher energy to a lower energy state and emits a gamma photon in the process. Gamma decay does not change the nuclear composition (number of protons and neutrons) but serves as a way to release excess energy from excited nuclei. The gamma ray is a photon with no charge and zero rest mass (travels at the speed of light), and its trajectory is unaffected by the presence of external electric and magnetic fields.

During beta decay, the atomic nucleus emits a beta particle—a high-speed electron (rest mass $m_0$ = 0.511 MeV/c$^2$)—that forms at the time of the decay event. The energies of the emitted beta particles form a continuous range of energies and therefore give rise to a continuous range of beta particle velocities. The energy range for beta particles depends on the particular nucleus and extends from 0 to some maximum energy. For example, $^{210}$Bi is a beta emitter that has a beta particle energy extending from 0 MeV up to a maximum energy of 1.2 MeV with an average around 0.2 MeV [10]. Table 1 lists the most important beta emitters relevant to the Chernobyl accident [11], along with the average and maximum energies of the beta particles emitted by these isotopes. The weighted average speed for beta particles listed in Table 1 is 0.63*c* (63% of the speed of light), while the average maximum speed of the isotopes listed in Table 1 is 0.83*c*.

An alpha particle (rest mass $m_0$ = 3727.379 MeV/$c^2$) is a helium nucleus, consisting of two protons and two neutrons. Unlike beta particles that have a distribution of energies, alpha particles are ejected with specific energies that depend on the identity of the radioactive isotope. For most isotopes, alpha particles have energies between 4 MeV and 9 MeV [12–14], corresponding to velocities of 0.05$c$ to 0.07$c$, respectively. Some important Chernobyl alpha emitters [11] are listed in Table 2 and collectively show an average speed of 0.052$c$.

According to the β formula in Equation (2), alpha particles of 4.9 to 6.1 MeV have speeds between 0.0512c and 0.0572c, while the speed of a 3.54 MeV electron (from Rh-106, Table 1) is as high as 0.992c. Our goal is to compare the radii of curvature in constant magnetic and electric fields of alpha and beta trajectories on the medal to show that the beta trajectory is depicted as too large, relative to the alpha particle.

**Table 1.** Important beta-emitting radioactive isotopes of the Chernobyl accident and their energies [13]. For isotopes with multiple beta decay branches, the values listed correspond to an average over the branches, each weighted according to the intensity of the branch. The particle speeds were calculated from the energy using Equations (1) and (2). * An additional short-lived daughter product of Ru-106 is Rh-106, with a maximum emitted beta particle energy of 3.54 MeV corresponding to a particle speed of 0.992$c$.

| Isotope | References | Emitted Particle Energy (MeV) | Particle speed ($c$ [1]) |
|---------|-----------|-------------------------------|--------------------------|
| Ba-140 | [15,16] | 0.281 (average), 0.828 (maximum) | 0.764$c$, 0.924$c$ |
| Ce-141 | [15,17,18] | 0.145 (average), 0.479 (maximum) | 0.627$c$, 0.856$c$ |
| Ce-144 | [15,19] | 0.082 (average), 0.290 (maximum) | 0.508$c$, 0.770$c$ |
| Cs-134 | [15,18,20] | 0.157 (average), 0.498 (maximum) | 0.644$c$, 0.862$c$ |
| Cs-136 | [15,21,22] | 0.118 (average), 0.396 (maximum) | 0.583$c$, 0.826$c$ |
| Cs-137 | [15,23,24] | 0.188 (average), 0.551 (maximum) | 0.682$c$, 0.877$c$ |
| I-131 | [15,25] | 0.182 (average), 0.578 (maximum) | 0.675$c$, 0.883$c$ |
| I-133 | [15,26,27] | 0.405 (average), 1.140 (maximum) | 0.830$c$, 0.951$c$ |
| Kr-85 | [13,15,28] | 0.251 (average), 0.685 (maximum) | 0.741$c$, 0.904$c$ |
| Mo-99 | [13,15,29] | 0.389 (average), 1.080 (maximum) | 0.823$c$, 0.947$c$ |
| Np-239 | [15,27,30] | 0.118 (average), 0.407 (maximum) | 0.583$c$, 0.831$c$ |
| Ru-103 | [15,31] | 0.064 (average), 0.224 (maximum) | 0.458$c$, 0.719$c$ |
| Ru-106 * | [15,32] | 0.010 (average), 0.039 (maximum) | 0.195$c$, 0.372$c$ |
| Sr-89 | [13,15,33] | 0.585 (average), 1.495 (maximum) | 0.885$c$, 0.967$c$ |
| Sr-90 | [15,23,34] | 0.196 (average), 0.546 (maximum) | 0.691$c$, 0.875$c$ |
| Te-132 | [15,35,36] | 0.067 (average), 0.240 (maximum) | 0.467$c$, 0.733$c$ |
| Xe-133 | [15,26,27] | 0.100 (average), 0.346 (maximum) | 0.549$c$, 0.803$c$ |
| Zr-95 | [15,37] | 0.117 (average), 0.388 (maximum) | 0.581$c$, 0.823$c$ |

[1] Where $c$ is the speed of light (in m/s).

**Table 2.** Important alpha-emitting radioactive isotopes of the Chernobyl accident and their corresponding alpha particle energies [18,23,27,38]. For isotopes with multiple alpha decay branches, the values listed correspond to the average overall branches weighted according to the intensity of the branch. The particle speeds were calculated from the energy using Equations (1) and (2).

| Isotope | Emitted Particle Energy (MeV) | Particle Speed ($c$ [1]) |
|---------|-------------------------------|--------------------------|
| Am-241 | 5.479 | 0.0542$c$ |
| Cm-242 | 6.101 | 0.0572$c$ |
| Pu-238 | 5.487 | 0.0542$c$ |
| Pu-239 | 5.236 | 0.0530$c$ |
| Pu-240 | 5.155 | 0.0526$c$ |
| Pu-242 | 4.892 | 0.0512$c$ |
| U-235 | 4.394 | 0.0485$c$ |
| U-238 | 4.187 | 0.0474$c$ |

[1] Where c is the speed of light (in m/s).

## 2. Analysis

### 2.1. The Lorentz Force—Charged Particle Dynamics in External Fields

The Lorentz force governs the motion of charged particles in external electric and magnetic fields and is used to determine the particle trajectories. The force experienced by a particle of charge $q$ with velocity $v$ in an external electric field $E$, and magnetic field $B$, is given by [9,39]:

$$\vec{F} = q\left(\vec{E} + \vec{v} \times \vec{B}\right) \tag{3}$$

For simplicity, our analysis will focus separately on the trajectories in either a static uniform magnetic field or a static uniform electric field, one of which is usually depicted in textbook figures when discussing the effect of charge and mass on radiation direction. The coordinate system adopted for this analysis is shown superimposed upon the central figure of the CLM (Figure 3). All radioactive emission is assumed to occur at the origin at time $t = 0$ and to initially be traveling along the positive $y$-axis. Also, all particle trajectories are calculated assuming the particles are in a vacuum.

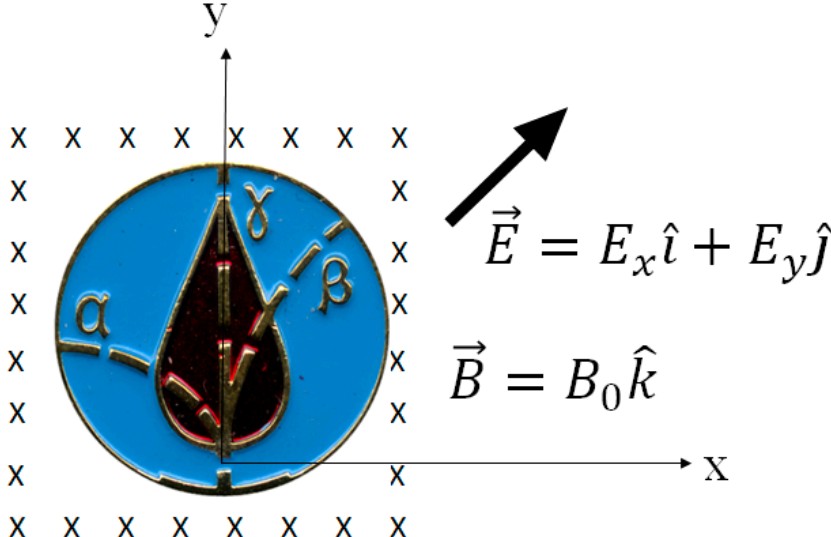

**Figure 3.** An enlarged view of the central figure of the liquidator medal, upon which our choice of coordinate system is superimposed. The origin is chosen between the bottom of the drop of blood depicted and the outer edge of the CLM. The positive $z$-axis points out of the page. Also shown are the orientation of a general electric field in the page with x and y components, $E_x$ and $E_y$, respectively, and a general magnetic field perpendicular to the plane of the page.

### 2.2. Relativistic Equations of Motion in a Static, Uniform Magnetic Field

For a static uniform magnetic field, Equation (3) becomes

$$\vec{F} = q\left(\vec{v} \times \vec{B}\right) \tag{4}$$

For a static, uniform magnetic field, the energy of the charged particle is constant in time and therefore, $v$, and $\gamma$ are both constant in time as well [9]. To be consistent with the CLM depiction, we will consider a magnetic field of magnitude $B_0$ directed into the page (along the negative $z$-axis), $\vec{B} = -B_0\hat{k}$. The resulting trajectory for a particle with our initial conditions can now be identified with a circle in the medal plane, with the gyroradius derived from equating the magnetic force F with the centripetal force:

$$F = qvB_0 = \frac{mv^2}{r} \text{ giving } r = \frac{mv}{qB_0} = \frac{\gamma m_0 v}{qB_0} \tag{5}$$

Using Equation (5), the ratio of radii between an alpha and a beta particle trajectory both traveling in the same magnetic field is

$$\frac{r_\alpha}{r_\beta} = \frac{m_{0\alpha}}{m_{0\beta}} \frac{q_\beta}{q_\alpha} \frac{\gamma_\alpha v_\alpha}{\gamma_\beta v_\beta} \tag{6}$$

The ratio of rest masses, $\frac{m_{0\alpha}}{m_{0\beta}} = 7294$, and charges, $\frac{q_\beta}{q_\alpha} = -\frac{1}{2}$, cannot be adjusted, so the particle velocities and gamma factors (which also depend on the velocity) are the only parameters that can be varied to set the trajectories. Inserting the minimum alpha speed of 0.0512c and the maximum beta speed of 0.992c into Equation (6), we have

$$\frac{\gamma_\alpha v_\alpha}{\gamma_\beta v_\beta} = \frac{\frac{0.0512c}{\sqrt{1-(0.0512)^2}}}{\frac{0.992c}{\sqrt{1-(0.992)^2}}} \approx \frac{1}{153} \tag{7}$$

which gives,

$$\frac{r_\alpha}{r_\beta} \approx -23.8 \tag{8}$$

The radius of the alpha particle trajectory should be 23.8 times larger than that of the beta particle. The minus sign accounts for the fact that the oppositely charged particles deflect in different directions. Therefore, we expect that the beta particle will be deflected much more in a given static uniform magnetic field than an alpha particle.

However, this result is in disagreement with the trajectories depicted on the CLM. The medal clearly shows a much larger deflection for the alpha particle relative to the beta particle deflection, when in reality it is the opposite, with the gyroradius of the alpha particle about 24 times larger than the largest beta trajectory. Numerous textbooks (covering many different science branches) and journal figures unrealistically exaggerate the angle of deflection to emphasize that the emitted species have opposite charges while not accounting for the effects of mass and speed [4–8].

*2.3. CLM Trajectory Analysis*

A simple quantitative analysis of the trajectories shown on the liquidator medal was carried out using circular orbits (Figure 4), which indicate that the ratio of radii is

$$\frac{r_\alpha^{CLM}}{r_\beta^{CLM}} \approx -\frac{1}{2.5} = -0.4 \tag{9}$$

This radii ratio would require that Equation (7) have

$$\frac{\gamma_\alpha v_\alpha}{\gamma_\beta v_\beta} = \frac{1}{9118} \tag{10}$$

If we take the alpha particle speed as 0.0512c, then to acquire the above gyroradius ratio the speed of the beta particle would have to be 0.999998c, corresponding to a beta particle energy of 255 MeV. Figure 5 shows the trajectories of alpha and beta particles, calculated according to Equation (5), in a static uniform magnetic field of magnitude 43 T directed into the page superimposed upon the central figure of the CLM. The magnetic field value used was determined to give a quantitative fit for the alpha particle trajectory (blue line) corresponding to a speed of 0.0512c. Also shown in the figure are the trajectories for a beta particle (red line) with initial velocity 0.992000c and a gamma ray with velocity c. The black line in Figure 5 was calculated from Equation (10) and corresponds to the beta velocity of 0.999998c needed to reproduce the beta particle trajectory depicted on the CLM.

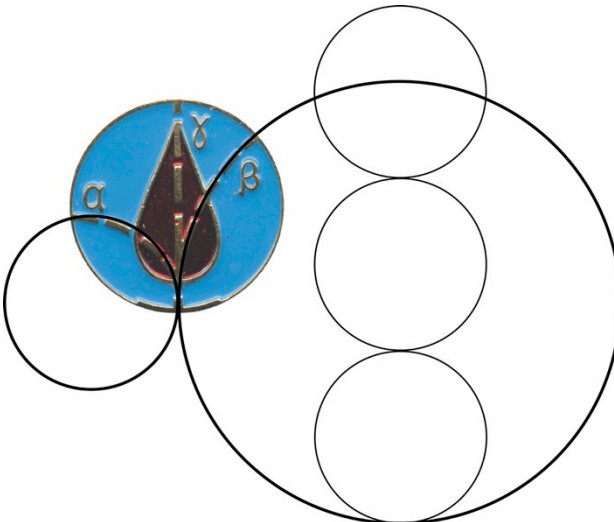

**Figure 4.** Semi-quantitative analysis of the alpha and beta particle trajectories depicted on the liquidator medal. The radius of the trajectory of the beta particle is approximately 2.5 times larger than the radius indicated for the alpha particle.

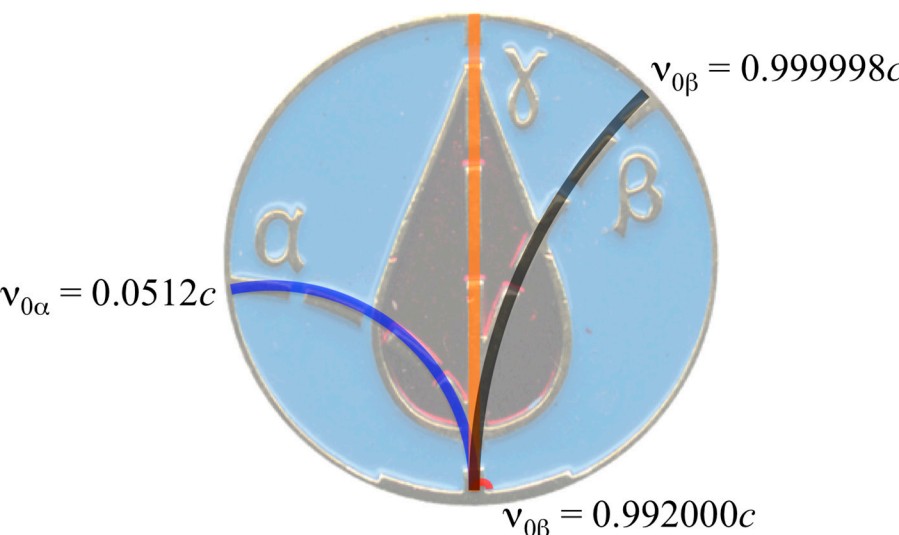

$v_{0\beta} = 0.999998c$

$v_{0\alpha} = 0.0512c$

$v_{0\beta} = 0.992000c$

**Figure 5.** Comparison of particle trajectories for a magnetic field of magnitude 43 T calculated using Equation (6) with the parameters mentioned in the text with the actual CLM depicted trajectories. The figure was generated by plotting the trajectories (assuming an alpha particle speed of $0.0512c$) with scaled axis and then superimposing an appropriately scaled CLM image behind the trajectories. Also shown for comparison is the trajectory of a beta particle with a speed of $0.992000c$.

Another possibility is to assume that the beta particle has a speed of $0.6c$ and that the alpha particle is traveling approximately 9118 times slower than the beta particle. This would correspond to an alpha particle traveling at a speed of $0.00008c$. However, this velocity is far too low for an alpha particle speed at the moment of emission. The only instance where this might occur is in a situation where the velocities of the alpha and beta particles have been modified to different degrees during passage through a barrier. This would require that the alpha particle's velocity be reduced substantially more than the beta particle's velocity. However, any barrier that would dramatically reduce the speed of an alpha particle, without absorbing its energy entirely, would also likely result in substantial deflection of the beta particle trajectories (i.e., such as scattering as seen in a cloud chamber that affects beta particles much more than alpha particles). Therefore, we would not expect a result similar to

Rutherford's original diagram, or the simplified trajectories depicted in textbooks [4–8] and on the Chernobyl Liquidator Medal.

### 2.4. Relativistic Motion of a Charged Particle in a Static Uniform Electric Field

For a charged particle in a static uniform electric field, Equation (3) becomes

$$\frac{dp_x}{dt} = qE_{0x} \text{ and } \frac{dp_y}{dt} = qE_{0y} \tag{11}$$

where $p$ is the particle momentum.

With the initial condition that the particle is traveling along the positive *y*-axis with momentum $p_0$, integration of Equations (11) yields

$$p_x = qE_{0x}t \text{ and } p_y = qE_{0y}t + p_0 \tag{12}$$

Using the relativistic Equations

$$E^2 = \left(m_oc^2\right)^2 + p^2c^2 \tag{13}$$

and

$$\vec{v} = \frac{\vec{p}\,c^2}{E} \tag{14}$$

In Equations (13) and (14), the total energy can be expressed as

$$E = \sqrt{(m_0c^2)^2 + \left[(qE_{0x}t)^2 + \left(qE_{0y}t + p_0\right)^2\right]c^2} \tag{15}$$

and the trajectories by

$$x(t) = \int_0^t \frac{(qE_{0x}t')c^2dt'}{\sqrt{(m_0c^2)^2 + \left[(qE_{0x}t')^2 + \left(qE_{0y}t' + p_0\right)^2\right]c^2}} \tag{16}$$

$$y(t) = \int_0^t \frac{\left(qE_{0y}t' + p_0\right)c^2dt'}{\sqrt{(m_0c^2)^2 + \left[(qE_{0x}t')^2 + \left(qE_{0y}t' + p_0\right)^2\right]c^2}} \tag{17}$$

Analytical solutions can be obtained from Equations (16) and (17), but we shall leave them in this form as they can easily be numerically integrated to determine the particle trajectory. Numerical integration was conducted using MAPLE 14 math software.

We first consider an electric field containing only an x component. For an alpha particle velocity of 0.0512$c$, the magnitude of the electric field that gives a qualitative fit was found to be $E_x = -19 \times 10^8$ N/C (equivalent to V/m). For the electric field analysis, the best fits were obtained by choosing the origin to be near the base of the blood drop rather than at the bottom of the CLM as was done for the magnetic field analysis. This was done to allow the best fit possible with the trajectories depicted on the medal. Figure 6 shows the trajectory of an alpha particle of speed 0.0512$c$ (blue line) along with the beta particle trajectory (red line) corresponding to a speed of 0.9920$c$. Like the magnetic field analysis, the beta particles should be deflected much greater than the alpha particle. To emulate the situation depicted on the CLM, assuming a static uniform electric field with $E_x = -19 \times 10^8$ N/C, a beta particle would require a velocity of 0.9998$c$, corresponding to a beta particle energy of 25 MeV.

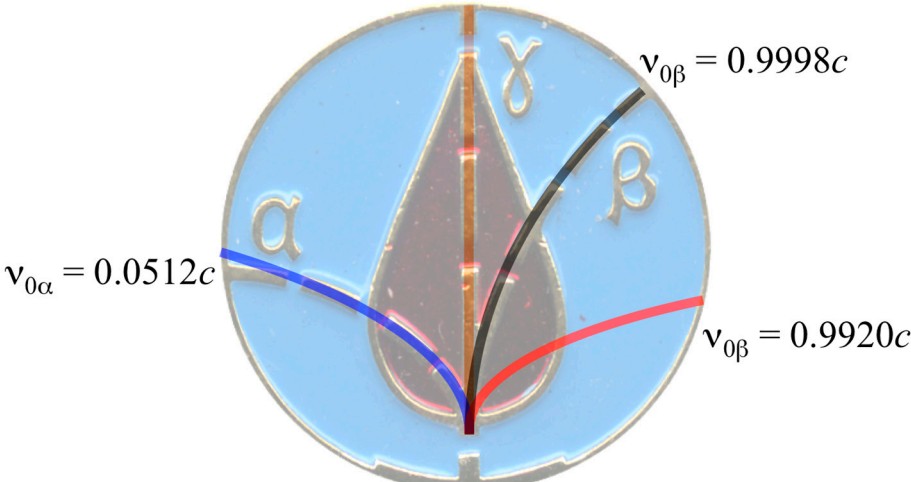

**Figure 6.** Comparison of particle trajectories calculated using Equations (16) and (17) utilizing a horizontal electric field of strength $E_x = -19 \times 10^8$ N/C with the actual CLM depicted trajectories. Also shown for comparison is the trajectory of a beta particle with speed of $0.9920c$.

An improved fit for the alpha trajectory can be found by using an electric field having both x and y components (Figure 7). Using an electric field with $E_x = -9.0 \times 10^8$ N/C and $E_y = -4.0 \times 10^8$ N/C, the alpha trajectory has a better fit (blue line) and the beta trajectory (red line) shows less of a deflection. Using this electric field, a beta particle of velocity 0.9985c (black line), corresponding to an energy of 29 MeV, would behave similarly to the trajectories depicted on the CLM. Shown for comparison is the trajectory for a beta particle with a velocity of $0.9920c$ (red line).

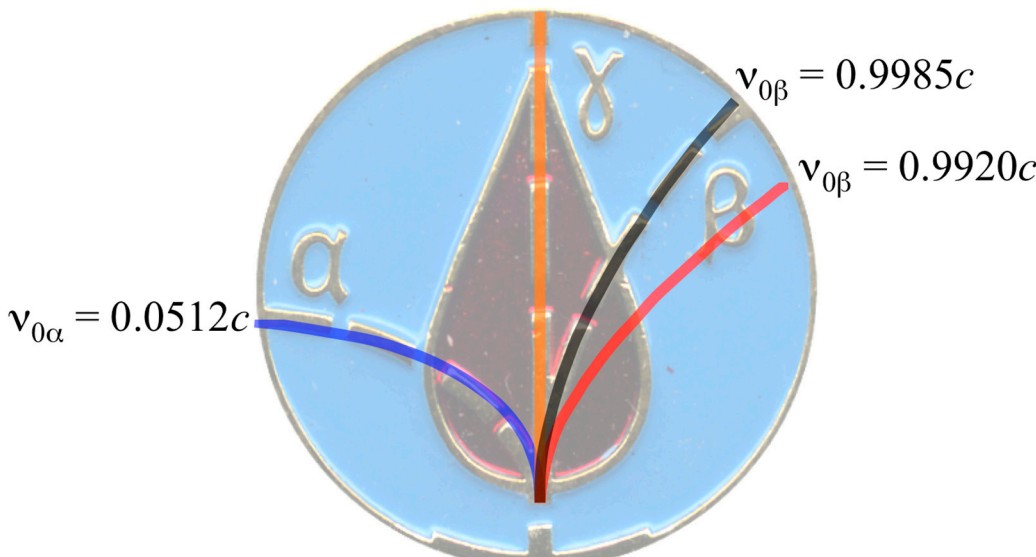

**Figure 7.** Same as Figure 6 except the electric field has components of $E_x = -9.0 \times 10^8$ N/C and $E_y = -4.0 \times 10^8$ N/C. These field values give an improved trajectory to the alpha particle and lower the beta particle speed needed to reproduce the trajectories depicted on the CLM.

## 3. Discussion

The parametric equations describing the relativistic trajectories of charged particles in a static uniform magnetic field and a static uniform electric field were used to analyze the trajectories that are depicted on the CLM.

For a static uniform magnetic field of −43 T, an alpha particle velocity of 0.0512*c* required a beta particle with a velocity of 0.999998*c*, corresponding to an energy of 255 MeV, to reproduce the trajectories depicted on the CLM. The maximum beta particle energies for most radioactive isotopes range from 0.01 MeV to 4 MeV [12], which corresponds to an upper speed of 0.9936*c*. The large velocity required to reproduce the scene on the CLM is well outside of beta particle energies observed during spontaneous nuclear decays and is only observed in particle accelerators such as the synchrotron, which can accelerate electrons above 1000 MeV [40].

In a static uniform electric field, the velocities of beta particles needed to reproduce the trajectories depicted on the CLM (when assuming an alpha velocity of 0.0512*c*) were found to be smaller than the velocities needed for a static uniform magnetic field. However, the beta particle velocities to reproduce the CLM figure in a static uniform electric field were still much higher than the typical beta velocities of the relevant radioisotopes listed in Table 1. An electric field with both x and y components was found to give a better fit to the CLM trajectories along with reduced beta particle velocities needed to reproduce the CLM trajectories, however, the reduced velocities still fell outside the range of the velocities in Table 1.

The trajectories depicted on the CLM do not correspond to realistic beta particle energies found for radioisotopes relevant to the Chernobyl accident within the assumption of a static uniform magnetic or electric field.

It is important to note that the field strengths presented above were modeled for a vacuum and were based upon the blue background on the medal having a diameter of 1.8 cm. In reality, these electric field strengths would be orders of magnitude greater than what could actually exist in an atmospheric setting because electrical discharge would preclude their formation (i.e., lightning) [41]. Varying the scale of the blue field (i.e., making it larger) would reduce the field strengths necessary to reproduce the trajectories. However, this produces its own set of problems in an atmospheric setting due to absorption or deflection of alpha and beta particles that would prevent a satisfactory match.

It is also interesting to note that many textbooks [4–8], when discussing the deflection of nuclear radiation in a magnetic/electric field, usually exaggerate the deflection of the alpha particles to emphasize the effect of charge on the trajectories. Could it be that the liquidator medal was modeled from an erroneously exaggerated textbook figure illustrating nuclear decay trajectories in an electric or magnetic field? This is consistent with the fact that our calculated trajectories, for reasonable particle velocities, closely match those depicted on the medal. For this reason, we believe that the most likely explanation is that the particle trajectories were mislabeled, with alpha and beta switched. Another possibility is that the radii were estimated by only considering the charge of the particles, not taking into account the effect of mass (assuming the mass of the beta and alpha particle were identical). Considering only the charges, Equation (8) would give a radii ratio of −0.5, which is close to the actual value of −0.4 depicted on the CLM.

## 4. Conclusions

We would like to acknowledge the work and dedication of the Chernobyl liquidators during the years following the accident. Examination of the α, β, and γ radiation trajectories depicted on the famous Chernobyl Liquidator Medal reveals an erroneous artistic rendering of Rutherford's famous diagram. Additionally, it is clear that scientific misconceptions (or alternative conceptions) can strongly impact student learning [42]. Renditions showing the deflection of nuclear decay products have necessarily permeated all of the sciences, with varying degrees of artistic license. We hope that our observations here will prompt others to reexamine the usage of this and other classic scientific figures to avoid problems and misconceptions in the future.

**Author Contributions:** J.L. developed the idea for the manuscript, and M.M. provided the mathematical analysis of particle trajectories. Both authors were involved in the writing of the manuscript and construction of figures.

**Funding:** This research received no external funding.

**Acknowledgments:** We would like to thank Steve Dutch for translating the medal's text and Jorge Estévez for reading and providing comments. Brian Welsch, Scott Ashmann, Michael Hencheck, Heidi Fencl, and two anonymous reviewers also provided valuable comments.

**Conflicts of Interest:** The authors declare no conflict of interest.

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
