# Peer review of "Chernobyl’s Lesser Known Design Flaw: The Chernobyl Liquidator Medal—An Educational Essay"

_2571-8800, doi:10.3390/j2030023_

Round 1
Reviewer 1 Report
First I thought it was a quite insignificant paper, but then I recognized its educational value and to make that clear I suggested an addition to the title. It is a good exercise in practical special relativity and mechanics and I like it. There are a few things that I wanted to point out, some parts can be shortened and there are also a few errors. You can find all my detailed views in the attached docx file. I also made an Excel sheet that the authors might find useful to look at and use. As there is no way, as I understand it, to attach two files I send the Excel sheet directly to Jerry Chen .
Best Regards,

Reviewer 2 Report
This paper is not a research paper but has some educational interest and a sociological interest.
I agree with author that most probably the alpha and beta labels have been erroneously switched in the medal.
The author has correctly shown that a single vertical B field cannot account for the trajectory.
The author has correctly shown that a single electrical E field in the xy plane cannot account for the trajectory.
If the purpose of this paper is educational, it is funny that the author does not recognize that a vertical B field combined with an electrical field in y direction is an easy solution.
This system can be analytically solved with some undergraduate level math (much simpler that the examples shown by the author)
If the editor agree for publication in this journal of such a paper,
I suggest the following major revision of the paper:
-remove the unnecessary and heavy/numerical treatment of the single fields cases, just commenting with few sentence that a solution is not existing in these cases.
-add a paragraph with this analytical solution and a picture superimposing the trajectory:
For author assistance I give here a simple plain text version of the solution:
The trajectory drawn in the medal can be solved through a constant (negative) electric field E in y direction and constant (negative) magnetic field B in z direction.
So both particle are deflected in opposite x direction by the magnetic field, beta particle is attracted in y direction by electric field and alpha particle is just repelled.
The motion information from medal drawing are:
a1) Vz = 0 at each time
a2) Vx(0) = 0 both particle are intially going in y direction
a3) after a time Ta the alpha particle has moved a quantity Ya
toward y direction and Vy(Ta) = 0 i.e. Px(Ta) = P(Ta)
(alpha particle is horizontally directed at medal border exit point)
a4) after a time Tb the beta particle has moved a quantity Yb
toward y direction and Vy(Tb) = Vx(Tb)
(beta particle is 45 degree inclined at medal border exit point)
a5) Yb = 2Ya
The motion equation in the two axis are:
eq1: dPx/dt = qBVy
eq2: dPy/dt = qE - qBVx
eq3: integrating eq1 for both particle, knowing Px(0)=0
one has: Px(t) = qBY(y)
Alpha particle motion (q=2 m=Ma):
use of NON relativistic approximation
momentum P = mV ; kinetic energy K = P^2/(2m)
and Ka = K(0) is the few MeV average alpha energy
the eq3 become: Px(Ta) = qBYa = P(Ta)
K(Ta) = (qBYa)^2/(2m)
use of Energy conservation: K(Ta) = Ka + qEYa
permits to relate E value to B value:
eq4: E = [K(Ta)-Ka]/qYa (it is negative)
Beta particle motion (q=-1 m=Mb):
Fully relativistic treatment
the eq3 become: Px(Tb) = qBYb = Py(Tb)
(using Vx(t)/Vy(t) = Px(t)/Py(t) @ Tb trajectry is at 45 degree)
therefore P(Tb) = sqrt(2)qBYb
use of Energy conservation:
eq5: sqrt(P^2(Tb)+m^2) - sqrt(P^2(0)+m^2) = qEYb (this is positive)
where one can rename sqrt(P^2(0)+m^2) = Kb+Mb
where Kb is the few hundred keV average kinetic energy of beta particle
substituting the electric field from eq4 in eq5 one has:
eq6: sqrt(2*B^2*Yb^2+Mb^2) - Mb -Kb = [Ka-4B^2Ya^2/2Ma]Yb/2Ya
one can use Yb = 2Ya = Y and renaming F=B^2Y^2 and Ktot = Ka+Kb
2F = (Mb+Ktot-F/2Ma)^2-Mb^2 = (Ktot-F/2Ma)*(Ktot-F/2Ma+2Mb)
a good approximation is to neglect F/2Ma wrt Ktot
in this way F = Ktot(Ktot+2Mb)/2 is the solution for B^2Y^2
(and it is true that Ktot >> F/2Ma = Ktot * (Ktot+2Mb)/4Ma)
Therefore also the B value is not fixed and is related
to the absolute (unknown) scale length Y.
I leave to the author the exercise to substitute the
numerical value of B as the Earth magnetic field,
and to find the Y scale length and to evaluate the
necessary electric field E and to comment if this
value could be a feasible value
and the effect of the crossed air.
I suggest to add some impressive sentence in memory
of Chernobyl Liquidators in the conclusions.
Author Response
Please see attached PDF file.

Round 2
Reviewer 1 Report
I have now read through the revised paper and checked how the authors have reacted on my suggestions. In my paper with suggestions I sent previously I have marked green the those, they have taken satisfactory action on.
White text against a red background spells out a few new comments.
As Pu-241 is used but shouldn't, there is a need for recalculation a few things. That will not create any important changes but looks better when people want to repeat the calculations, especially in a school situation.
Maybe somewhere where the unit N/Q is first used they should point out that it is the same as V/m.
Just at the end of the review process this time I started to think that the electric fields are so enormous (up to 1330 MV/m) when 3 MV/m is enough to break through the air like in thunder. If there is no miscalculation that would be enough to say that no realistic electrical field would be enough to reproduce the trajectories.

Author Response
Responses to Reviewer #1
Reviewer #1 had comments in a Word file, as well as in the Review Report Form. We will address the Word file comments first.
Line 49: This comment prompted us to double check our measurement of the diameter of the blue circle in the medal. Although the larger circle is nearly 3 cm, the blue circle itself is only 1.8 cm. This prompted us to recalculate our field strengths and beta particle speeds. This did not significantly change the outcome or conclusions of the paper. We added the diameter to the figure caption, as requested.
Line 108: This appears to be the only major place where we disagree with the reviewer. We leave this decision up to the editor. We feel that citing the individual sources recognizes those authors in a way similar to a special volume containing individual research papers would. It is appropriate to cite the papers themselves, not the editor(s) of the overall volume.
The reviewer wrote, “if the authors want to do this let them!”, so we would prefer to keep the citations as they are, but would change them if the editor feels it is necessary.
Line 109: Changes made, as requested.
Lines 109-122 (comments about Cm-244 and Pu-241): We agree with the reviewer and have removed Pu-241 from the table, as requested. However, because Pu-242 still remains as the lowest energy artificial isotope (velocity of 0.0512c), our alpha particle values did not change because of this. Pu-242 is a 100% alpha emitter, so it’s appropriate to keep in the table.
Lines 324-334: The author color coded this as green, so we think this issue is resolved, but understand that they were expressing their opinion in red. Please let us know if the editor feels we should further modify the paragraph.
Reviewer #1 Comments on the Review Report Form page:
Pu-241 comment was addressed above.
N/Q or N/C comment to add V/m in the text: We agree and made this change in the text at the first instance of N/C.
Final comment on enormous electric fields: We agree with the reviewer and have added a paragraph to the discussion (with a new reference) to help explain some of this to the reader. We hope our explanation is satisfactory.
Error Corrections:
Two small errors were discovered during this review that do not substantively alter the results of the paper. One was prompted by a comment from Reviewer #1 requesting the addition of the diameter of the blue circle on the medal. Because the diameter was 1.8 cm instead of 3 cm, we adjusted our field strengths and beta velocities appropriately.
In addition, we noticed that there was a small error in equation 16 (E0x should have been E0y in Equation 16b). We made this correction as well.
Reviewer 2 Report
I think the paper fulfill the journal requirements in the current version.
Author Response
The reviewer did not have any changes requested during review round 2. We thank the reviewer for their careful attention to the manuscript.